# Toothbrushes as a Source of DNA for Gender and Human Identification—A Systematic Review

**DOI:** 10.3390/ijerph182111182

**Published:** 2021-10-25

**Authors:** Govindarajan Sujatha, Veeraraghavan Vishnu Priya, Alok Dubey, Sheetal Mujoo, Ayman M. Sulimany, Ali Mohammed Omar Tawhari, Lujain Khalawi Mokli, Arwa Jaber Mohana, Saranya Varadarajan, Thodur Madapusi Balaji, A. Thirumal Raj, Shankargouda Patil

**Affiliations:** 1Department of Oral Pathology and Microbiology, Sri Venkateswara Dental College and Hospital, Chennai 600130, India; drgpsujatha@gmail.com (G.S.); vsaranya87@gmail.com (S.V.); thirumalraj666@gmail.com (A.T.R.); 2Saveetha Institute of Medical and Technical Sciences, Research Scholar Saveetha Dental College, Saveetha University, Chennai 600077, India; 3Department of Biochemistry, Saveetha Institute of Medical and Technical Sciences, Saveetha Dental College, Saveetha University, Chennai 600077, India; drvishnupriyav@gmail.com; 4Department of Preventive Dental Sciences, College of Dentistry, Jazan University, Jazan 45142, Saudi Arabia; dentaalok@yahoo.com; 5Division of Oral Medicine and Radiology, College of Dentistry, Jazan University, Jazan 45142, Saudi Arabia; sheetalmujoo@yahoo.co.uk; 6Department of Pediatric Dentistry and Orthodontics, College of Dentistry, King Saud University, Riyadh 11545, Saudi Arabia; asulimany@ksu.edu.sa; 7College of Dentistry, Jazan University, Jazan 45142, Saudi Arabia; ali.tohary1417@gmail.com (A.M.O.T.); Lojainmokli@gmail.com (L.K.M.); ArwaJ.Mohanna@gmail.com (A.J.M.); 8Tagore Dental College and Hospital, Chennai 600077, India; tmbala81@gmail.com; 9Department of Maxillofacial Surgery and Diagnostic Sciences, Division of Oral Pathology, College of Dentistry, Jazan University, Jazan 45142, Saudi Arabia

**Keywords:** DNA, forensic odontology, gender identification, human identification, toothbrush

## Abstract

Background: Few studies have reported the use of toothbrushes as a reliable source of DNA for human or gender identification. The present systematic review with the available information was conducted to answer the focus question “Is a toothbrush a reliable source of DNA for human or gender identification?”. Methods: The keyword combination “Toothbrush” and “DNA” was used to search databases including MEDLINE, Scopus, and Web of Science along with a manual search of reference lists of relevant articles. Duplicates and irrelevant articles were excluded, and the remaining articles were fully read for the final selection of articles. The risk of bias of the included studies was evaluated using the Appraisal tool for Cross-Sectional Studies (AXIS tool). Results: Of the 130 articles obtained, 122 duplicates or irrelevant articles were eliminated. Following the full-text reading of eight articles, five articles were selected based on eligibility criteria. The five studies reported that a toothbrush is a good source of DNA irrespective of the time interval. In a few studies some samples were not sufficient for complete DNA profiling due to factors such as the method of DNA extraction. Conclusion: Although a toothbrush is an excellent source of DNA for human and gender identification, future studies with a larger sample size, appropriate control group, and standardized technique of DNA extraction need to be conducted. Additionally, factors influencing the quantity and quality of DNA in toothbrushes need to be determined with standardized techniques.

## 1. Introduction

Although several advances have taken place in science and technology, there has been little change in the occurrence of natural calamities and crimes [1]. Hence, the identification of human remains is vital for several reasons. The field of forensic medicine is endowed with the responsibility of identifying human remains after events like murders, accidents, calamities, and war. Human identification in such events is essential to investigate the nature of the calamity/accident and persons involved, assist the police and courts of law, and give closure to relatives. The common methods of human identification in forensic analysis include dermal ridge fingerprint [2] and radiological investigations [3], which cannot be used during natural calamities as these sources are susceptible to damage during disasters [4]. Dental remains such as restorations and anatomy are utilized for identification because they are a cost-effective, reliable method, and they are not destroyed in case of burns and severe trauma as teeth are resistant to desiccation, fire, and decomposition [5]. The conventional method of human identification is the comparison of post mortem and antemortem records. DNA profiling is also a technique that has been effective since its introduction in 1985 [6,7]. DNA is the basic genetic material that forms an essential part of the human genome and that carries information to manufacture, assemble, and maintain all the components of a living organism. It is chemically denoted as deoxyribonucleic acid with a double-stranded structure. Several DNA nucleotides are arranged linearly to form a polymer chain of a DNA molecule. Two such stands of DNA are bound by a non-covalent bond to form a double helix. The structural subunit of the DNA molecule is termed a nucleoside that has a backbone, sugar, and base. The genetic information is carried out by the four bases: adenine (A), cytosine (C), guanine (G), and thymine (T) [8]. DNA is characteristic of an individual and hence serves as an important tool for human identification. Genetic and epigenetic alterations also serve as markers for the diagnosis and prognosis of various diseases. For instance, Mansueto et al. have reported that higher levels of cell-free DNA demonstrate the extent of cellular damage, and circulating mitochondrial DNA is a marker of poor prognosis in patients with heart failure [9]. DNA is one of the most stable natural molecules that can be preserved for millions of years without losing structure and stability and can be used for human identification. Due to these properties, DNA is an important analytical tool in forensic analysis [10,11]. The sources of DNA include soft tissues such as human skin [12], dental pulp odontoblastic processes [13], and cellular cementum [14]. Body fluids that are rich sources of DNA include human saliva, which contains desquamated epithelial cells and can be obtained in a non-invasive manner [15]. Other external sources of human DNA include saliva-stained stamps, cigarette butts [16], toothbrushes, and oral prostheses [17]. Of all the external sources, the toothbrush is a reliable source of DNA that is easy to procure [6]. Moreover, the used toothbrush of the direct missing person is better than indirect reference samples from family members [18]. There are numerous factors that influence the successful collection and extraction of DNA. They are the duration of use of the toothbrush, the need for absolute certainty that the toothbrush belonged to the person being investigated and that the toothbrush has not been contaminated by others including members of the household, proper protocol in collection and transportation to obtain cellular material from the used toothbrush method, DNA extraction, and analysis of the evidence. 

The various known methods to obtain cellular content from a toothbrush include removing the toothbrush head with a hot scalpel and agitating the brush head with lysis solution, removing bundles of bristles followed by immersion in a lysis solution, cutting bundles of bristles from the brush head, and utilizing swabs proximal to the bristle base [19,20]. However, very few studies have been conducted to assess the reliability of toothbrushes as a source of DNA for human or gender identification. On the contrary, a study in Thailand failed to generate accurate results, possibly due to the temperature and humidity accelerating microbial growth that facilitates DNA degradation [21]. Additionally, the higher quantity of PCR inhibitors of toothpaste residues could affect the results. There are lacunae in knowledge on the various factors that affect the quantity and quality of DNA from used toothbrushes. With the available information, this systematic review was conducted to determine if the toothbrush is a reliable source of DNA.

## 2. Materials and Methods

Protocol and registration: The authors searched the international prospective register of systematic reviews (PROSPERO) to determine if there were registered protocols on a systematic review that reported the quantity or quality of DNA from toothbrushes. There were no such registered protocols obtained. The report of this systematic review was made according to the recommendations of the Preferred Reporting Items for Systematic Reviews and Meta-Analyses (PRISMA) statement [22,23].

Eligibility criteria:

Inclusion criteria: Studies were included when the following general criteria were met: (1)Original research articles that have assessed and quantified DNA and/or genes from toothbrush samples;(2)Original studies that have compared the quantity or quality of DNA from toothbrushes at various time intervals;(3)Original studies that have compared the quantity or quality of DNA from different methods of DNA extraction;(4)Study design was not a case report, editorial, letter to the editor, or review;(5)The report was published in the English language.

Exclusion criteria: Studies that did not assess or quantify DNA from toothbrush samples used by individuals or patients, and study designs that were case reports, editorials, letters to the editor, or review articles were excluded from the study.

Focus question: This systematic review aimed to address the potential focus question based on the following criteria: population (P): individuals who used the provided used toothbrushes for a specific time; study design (S): observational cross-sectional studies; and outcome (O): presence of DNA for human or gender identification. 

The focus question was “Is a toothbrush a reliable source of DNA for human or gender identification?”.

Search strategy: The keyword combination “Toothbrush” and “DNA” was used to search databases such as MEDLINE (accessed from PubMed), Scopus, and Web of Science on 18 June 2021. In addition, the authors manually scanned the reference lists of the included studies or relevant reviews identified through the search to ensure literature saturation. 

Study selection and data extraction: The study selection was conducted in three phases. Authors S.G. and S.V. participated through each phase of the review independently (screening, eligibility, and inclusion). Authors S.G. and S.V. independently screened the titles and abstracts for the elimination of irrelevant and duplicate articles. The full text of the remaining articles was screened by authors S.G. and S.V. to decide whether these meet the inclusion criteria. Articles on the detection and quantification of DNA from toothbrush samples were included in the present analysis. Neither of the authors was blind to the journal titles or the study authors or institutions.

Two reviewers (S.G. and T.M.B.) independently collected data on the study characteristics (author, year of study, and country), study design, sample size, study groups, type of toothbrushes, duration of use of the toothbrushes, method of DNA extraction, gene(s) assessed, overall results, inference, and statistical significance.

Risk of bias evaluation: The quality of the studies was evaluated through the Appraisal tool for Cross-Sectional Studies (AXIS tool) [24,25].

## 3. Results

Study selection: Of 130 articles obtained (64 from PubMed, 27 from Scopus, 37 from Web of Science, and 2 from cross-references), 122 were eliminated from the title and abstract screening as they were duplicates or irrelevant. Out of the 60 articles in PubMed only 4 were relevant to the study, and the rest of the articles had not aimed to assess human DNA from toothbrushes. Out of the 27 articles from Scopus only 1 article was relevant, 10 were duplicates, and the remaining 16 were irrelevant to the topic of interest. Out of the 37 articles from Web of Science 1 was relevant, 10 were duplicates, and the remaining 26 were not relevant to the topic of interest. The reason for so many articles not being relevant to the topic of interest may be attributed to the decreased number of articles in the present topic. Two articles were obtained from a cross reference of the relevant articles. Following the full-text screening of eight articles only five articles met the eligibility criteria and were included in this review. Two articles were rejected as they assessed the buccal mucosal cells that were obtained as a result of brush biopsy and one was a case report. The inter-examiner degree of agreement (Kappa) was 100% in the first stage (title and abstract screening stage) as well as in the second stage (eligibility and inclusion stage) of the study. The study selection process is depicted in Figure 1.

Study characteristics: Of the five studies included, one was from Saudi Arabia [26], one from India [27], one from Canada [28], one from Japan [19], and one from Thailand [20]. All studies used toothbrush samples of varying durations of use ranging from 1 day to 1 year. All the studies compared the quantity and quality of DNA from used toothbrushes at different durations. The study characteristics and data extraction are depicted in Table 1.

Risk of bias: None of the included studies provided adequate information on the methodology for statistical analysis, justification for sample size or sample frame, nor whether the selected samples were representative of the population. Of the included studies, three studies were graded as low risk of bias [26,27,28] and two studies were graded as moderate risk of bias [19,20], as these two studies had not conducted a statistical analysis. The results are depicted in Table 2.

Detailed description of the included studies: Among the five included studies that quantified DNA, one study assessed the SRY gene for gender identification [27], one study assessed DNA profiling using the Genetic Analyzer [26], one study assessed DNA profiling with a reference standard [28], and two studies assessed the STRs [19,20]. In all the studies DNA was present irrespective of the duration of use of the toothbrush [19,20,26,27,28]. However, complete profiling could be found in some samples and partial profiling could be attained in some samples. Only in one study, by Bandhaya et al., 2007 [20], were the samples were typed using the Chelex1-100 kits for DNA extraction.

Alfadaly et al., 2016 compared the DNA quantity and quality from siwaks and toothbrushes over different periods and reported that siwaks were a better source of DNA than toothbrushes [26]. Riemer et al., 2012 [28] did a partial head sampling of the used toothbrush for DNA extraction and reported that there was no significant difference in the quantity and quality of DNA obtained from 1 month, 3 months, and randomly used samples. Similarly, Tanaka et al., 2000 [19] reported complete DNA profiling for samples used for up to 1 year. Bandhaya et al., 2007 [20] compared the DNA content by using two techniques and using 5 and 10 bristle bundles from toothbrushes and reported that the DNA Mini Kit was a better method for DNA extraction and five bristle bundle toothbrushes are better suited for DNA extraction from toothbrushes. Considering gender identification, only one study, by Reddy et al., 2011, had assessed gender using the SRY gene, and reported a sensitivity of 100% and specificity of 73.33% [27]. 

## 4. Discussion

We reviewed the currently available scientific data on the usefulness of toothbrushes as a source of DNA for human or gender identification. Due to a lack of literature, only five articles that met the inclusion criteria were included in the present review. Considering the study characteristics, each study was from a different country with different populations and ethnic groups [19,20,26,27,28]. All the studies had employed a different technique for DNA extraction; hence the data obtained was heterogeneous. However, DNA was present in all the studies irrespective of the duration of use of the toothbrush, thereby endorsing the use of toothbrushes as an excellent source of DNA [19,20,26,27,28]. However, siwak—an Arabic word meaning tooth-cleaning stick, a type of ancient and traditional oral hygiene aid used in Arabian countries—was found to be a better source of DNA [26]. This could be attributed to the fact that siwak does not have PCR inhibitors, and does not contain toothpaste that can interfere with DNA yield. Additionally, the higher quantity of saliva and the inherent antimicrobial property of the natural herbal stick could prevent DNA degradation and increase the DNA yield. Considering the number of bristles used for DNA extraction, Bandhaya et al., 2007 [20], reported that 5 bristle bundles were a better source than 10 bristle bundles. This could be attributed to the fact that with a greater number of bristles, there will be a higher content of toothpaste that increases the chance of DNA degradation. The results are similar to that reported by Riemer et al., 2012 [28], who did a partial head sampling for DNA extraction and reported that there was no significant difference in the quantity and quality of DNA. The technique used for DNA extraction also has a major role in the quantity and quality of DNA extraction [20]. 

Regarding toothbrushes as a source of gender identification, there was only one study (with a low sample size of 30), which reported a 100% sensitivity of the SRY gene and 73% specificity [27]. The results have to be viewed with caution as although the study had a low risk of bias, the sample size, sample frame, and representativeness of the study population were not mentioned. Future studies with much larger samples need to be conducted to determine the exact usefulness of toothbrush samples for gender identification.

Considering that toothbrushes or siwaks are important sources of forensic evidence, it would be worthwhile to form protocols for the handling and storage of toothbrushes or siwaks as they could aid in critical problem solving in criminal proceedings. The studies included in this systematic review have revealed significant information about the recovery of DNA from toothbrushes, however it is to be reiterated that only a low yield of DNA could be harvested from the toothbrushes due to the fact that toothpastes contain an array of PCR inhibitors. We therefore recommend research and development towards the development of dentifrices free of PCR inhibitors considering the mammoth importance of toothbrushes from the forensic viewpoint. Moreover, the storage of toothbrushes in bathrooms is another concerning fact as toothbrushes could be contaminated by fecal plumes which also contain PCR inhibitors and would contaminate the toothbrushes with microbial and viral DNA. Hence, we recommend storing toothbrushes in a protected space.

One of the major limitations of the included studies is that none of the studies mentioned the chemical composition of the bristles, which could also affect cell adhesion and therefore could alter DNA yield [19,20,26,27,28]. There is also a lack of information on which part of the toothbrush is a good source of DNA. Hence, future studies need to report the chemical composition of the bristles of the toothbrush. Comparative studies on the DNA yield with different chemical compositions and DNA yield with different toothbrush parts could also be conducted.

## 5. Conclusions

Although all the studies had a moderate to low risk of bias, the sample size, statistical analysis used, sample frame, and representativeness of the samples were not mentioned. Moreover, none of the studies had a control group (unused toothbrushes kept in the same environment as that of the used toothbrush for the specific duration of time) nor mentioned the chemical composition of the bristles. However, with the limitations of the study all the toothbrush samples had DNA, although a few samples were not sufficient for complete DNA profiling. Irrespective of the duration of use, it can be concluded that toothbrushes are an excellent source of DNA. Future studies need to be conducted with a larger sample size, appropriate control group, and standardized techniques of DNA extraction to determine the usefulness of toothbrushes for human and gender identification.

## Figures and Tables

**Figure 1 ijerph-18-11182-f001:**
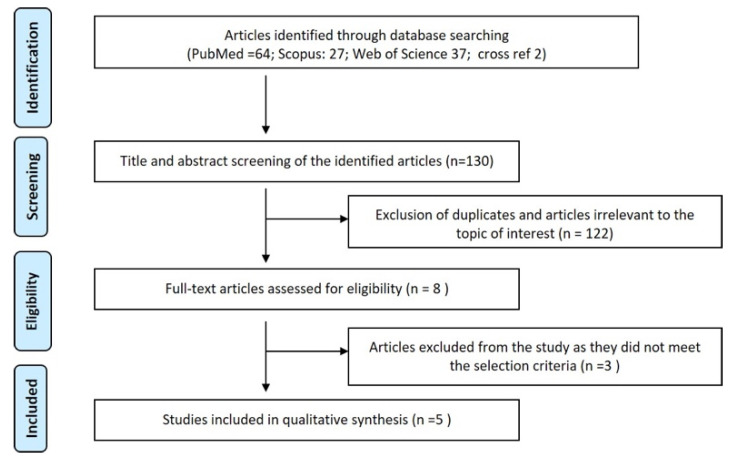
PRISMA flowchart.

**Table 1 ijerph-18-11182-t001:** Characteristics and detailed description of the included studies.

S. No.	Author Name/Year/Country	Study Design	Sample Size	Study Groups	Type of Toothbrushes	Duration of Use of Toothbrushes	Method of DNA Extraction	Gene(s) Assessed with Amplification Method	Capillary Electrophoresis Instrument	Results	Inference/Statistical Significance
1.	Alfadaly et al./2016/Saudi Arabia	Observational cross-sectional study	Total number of patients: 25; Total number of toothbrushes: 104; (25 uncovered siwaks, 4 covered siwaks, and 25 toothbrushes)	Group I (4 months, 4 volunteers); Group II (3 months, 5 volunteers); Group III (2 months, 4 volunteers); Group IV (1 month, 5 volunteers); Group V (1 week, 3 volunteers); Group VI (the same day, 4 volunteers, and covered siwak was added); Group VII (reference samples); Group VIII (positive and negative control samples)	Toothbrush, uncovered siwak, and covered siwak	1 week, 1 month, 2 months, 3 months, and 4 months	Promega kit	DNA profiling: AmpFLSTR Identifiler PCR Amplification Kit	3130XL Genetic Analyzer and GeneMapper software	DNA from siwak samples (18.96 ± 16.15) was higher than toothbrush samples (1.76 ± 1.07). 1/25 samples in siwak only partial DNA profiling could be done. In toothbrush group 8/25 profiling could be done and 3/25 partial profiling was done. Unit of DNA measurement was ng/μL	Significant (*p* < 0.001%)
2.	Reddy et al./2011/India	Observational cross-sectional study	30 samples, 30 patients (each used a sample for a week). Equal gender distribution in each group	Group 1: 10 samples immediately processed; Group II: 10 samples processed after 1 month; Group III: 10 samples processed after 2 months	Toothbrush brand name not mentioned	Immediately, 1 month, and 2 months	Real Genomics YGB 100 (Real Biotech Corporation, Taiwan) DNA extraction kit	*SRY* gene for gender identification using real-time PCR and Taq PCR Master Mix (Qiagen, India)	Real Plex Master Cycler (Eppendorf, Japan)	Genetic material was preset in low quantity in most of the samples. Gender identification: All males were identified correctly; out of 15 females, 4 were wrongly identified. Unit of DNA measurement was ng/μL	Sensitivity of SRY gene was 100%. Specificity of SRY gene in gender was 73.33%
3.	Riemer et al./2012/Canada	Observational cross-sectional study	N = 55 (25 males and 30 females)	Group I: 21 individuals used their toothbrush for 1 month; Group II: 22 individuals used their toothbrush for 3 months; Group II: 12 individuals gave their currently used toothbrush for analysis. Negative control: 2 unused toothbrushes	Toothbrush brand name not mentioned	Current used toothbrush, 1 month, and 3 months	Partial toothbrush head sampling technique phenol-chloroform method, and profiles were obtained using AmpFISTR Profiler Plus (Applied Biosystems, Foster City, CA)	DNA profile compared with reference standard AmpFISTR Profiler Plus PCR Identification Kit (Applied Biosystems, Foster City, CA)	No data available	DNA yield: 600 ng of DNA per toothbrush. DNA profiling complete in 51 and partial in 4 samples. Unit of DNA measurement was ng/g	No significant difference in the quantity and quality of DNA obtained from 1 month, 3 months, and random period used toothbrushes
4.	Tanaka et al./2000/Japan	Observational cross-sectional study	10 toothbrushes from 10 individuals	Case toothbrush samples (among 10 samples, 3 were actual cases: 1 drowned patient and 2 from murder case). Control: blood samples from 8 patients (blood samples from the heart of both the drowning and homicide victims were obtained at autopsy)	Used toothbrush	3 months to 1 year. Among 10 samples, 1 was stored for 6 months before analysis	Whole toothbrush head was taken for DNA extraction by phenolchloroform extraction and ethanol precipitation. Quantification by fluorometry	Six loci (DQA1, LDLR, GYPA, HBGG, D7S8, and GC); Nine STR loci (D3S1358, vWA, FGA, TH01, TPOX, CSF1PO, D5S818, D13S317, and D7S820). AmpFISTR Profiler Plus PCR Amplification Kit (Applied Biosystems, Foster City, CA)	310 Genetic Analyzer (Applied Biosystems, Foster City, CA)	9 toothbrushes: 10 to 430 ng; 10 toothbrushes: 0.5 ng mL. All loci were types in all samples despite low DNA yield. Unit of DNA measurement was ng/μL	All the test samples were typed at all loci. (No statistical analysis was performed)
5.	Bandhaya et al./2007/Thailand	Observational cross-sectional study	Total samples obtained from 4 individuals who used toothbrushes for 1, 7, 14, or 30 days	Group 1: toothbrush used for 1 day; Group 2: toothbrush used for 7 days; Group 3: toothbrush used for 14 days; Group 4: toothbrush used for 30 days. Each group had the following subgroups: Sub group a1: 5 bristle bundles DNA extraction done by Chelex1-100; Sub group a2: 5 bristle bundles DNA extraction done by QIAamp DNA Mini Kit; Subgroup b1: 10 bristle bundles DNA extraction done by Chelex1-100; Sub group b2: 5 bristle bundles DNA extraction done by QIAamp DNA Mini Kit	Used toothbrushes	1, 7, 14, and 30 days	Chelex1-100 or QIAamp DNA Mini Kit	STR (GeneAmp PCR System 9700) (Applied Biosystems)	ABI PRISM 3100 Genetic Analyzer and GeneMapper ID Software (Applied Biosystems)	DNA from QIAamp DNA Mini Kit > DNA from Chelex1-100 kit. Complete profile could be typed from QIAamp DNA Mini Kit but not in Chelex1-100 kit. Comparing 5 and 10 bristle bundles, STR was complete in all samples of Mini Kit from 5 bristle bundles and was complete in 30- and 14-day samples of 10 bristle bundles of Mini Kit. In the Chelex1-100 kit, none was completely typed. Unit of DNA measurement was ng/μL	Statistical analysis was not done. DNA Mini Kit was a better method for DNA extraction and 5 bristle bundles are better suited for DNA extraction from toothbrushes

**Table 2 ijerph-18-11182-t002:** Risk of bias of the included studies.

S. No.	Checklist	Alfadaly et al./2016/Saudi Arabia	Reddy et al./2011/India	Riemer et al./2012/Canada	Tanaka et al./2000/Japan	Bandhaya et al./2007/Thailand
1	Were the aims/objectives of the study clear?	Yes	Yes	Yes	Yes	Yes
2	Was the study design appropriate for the stated aim(s)?	Yes	Yes	Yes	Yes	Yes
3	Was the sample size justified?	No	No	No	No	No
4	Was the target/reference population clearly defined? (Is it clear who the research was about?)	Yes	Yes	Yes	Yes	Yes
5	Was the sample frame taken from an appropriate population base so that it closely represented the target/reference population under investigation?	Unclear	Unclear	Unclear	Unclear	Unclear
6	Was the selection process likely to select subjects/participants that were representative of the target/reference population under investigation?	No	No	No	No	No
7	Were measures undertaken to address and categorize non-responders?	Yes	Yes	Yes	Yes	Yes
8	Were the risk factor and outcome variables measured appropriate to the aims of the study?	Not applicable	Not applicable	Not applicable	Not applicable	Not applicable
9	Were the risk factor and outcome variables measured correctly using instruments/measurements that had been trialed, piloted, or published previously?	Not applicable	Not applicable	Not applicable	Not applicable	Not applicable
10	Is it clear what was used to determined statistical significance and/or precision estimates? (e.g., *p*-values, CIs)	Yes	Yes	Yes	No	No
11	Were the methods (including statistical methods) sufficiently described to enable them to be repeated?	No (statistical analysis was not described)	No (statistical analysis was not described)	No (statistical analysis was not described)	No (statistical analysis was not described)	No (statistical analysis was not described)
12	Were the basic data adequately described?	Yes	Yes	Yes	Yes	Yes
13	Does the response rate raise concerns about non-response bias?	Yes	Yes	Yes	Yes	Yes
14	If appropriate, was information about non-responders described?	Yes	Yes	Yes	Yes	Yes
15	Were the results internally consistent?	Yes	Yes	Yes	Yes	Yes
16	Were the results for the analyses described in the methods, presented?	Yes	Yes	Yes	Yes	Yes
17	Overall risk of bias	Low	Low	Low	Moderate	Moderate

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
