# Peer review of "Toothbrushes as a Source of DNA for Gender and Human Identification—A Systematic Review"

_ijerph, 2021, doi:10.3390/ijerph182111182_

Round 1
Reviewer 1 Report
This review shows that toothbrushes or siwaks are a possible source of DNA for person identification. Still, we need interlaboratory studies about DNA stability and recovery for its acceptance in court proceedings.

Author Response
Reviewer 1:
This review shows that toothbrushes or siwaks are a possible source of DNA for person identification. Still, we need interlaboratory studies about DNA stability and recovery for its acceptance in court proceedings.
Query: The authors may wish to add to the systematic review a hypothesis of future
interlaboratory study about toothbrushes or siwaks storage procedures for DNA recovery.
The analysis could be conduct with a standard multiplex to unify the process and start
implementation in forensic systems.
Response: The same has been included and highlighted
Considering that tooth brushes or Siwaks are an important source of forensic evidence it would be worthwhile to form protocols for handling and storage of tooth brushes tooth brushes or Siwaks as they could aid in critical problem solving in criminal proceedings. The studies included in this systematic review have revealed significant information about recovery of DNA from toothbrushes however it is to be reiterated that only low yield of DNA could be harvested from the tooth brushes due to the fact that toothpastes contain an array of PCR inhibitors. We therefore recommend research and development towards development of dentifrices free of PCR inhibitors considering the mammoth importance of toothbrushes in the forensic view point. Moreover, storage of tooth brushes in toilets is another concerning fact as tooth brushes could be contaminated by faecal plumes which also contain PCR inhibitors and would contaminate the tooth brushes with microbial and viral DNA. Hence we recommend storage of tooth brushes in a secluded space.
Minor Issues
Query: Line 55: Words "reason behind" could be erased
Response : The sentence has been rephrased and highlighted
Human identification in such events is essential to investigate the nature of the calamity/accident, persons involved, assist the police and courts of law and give closure to relatives
The authors should review punctuation, e.g. at lines 67, 68, 70, 74
Query: Line 67: Need to replace the v with a full stop after the word stability
Response : It has been done and highlighted
It has a double-stranded structure and is one of the most stable natural molecules that can be preserved for millions of years without losing structure and stability.
Query: Line 68: Need to replace the “(“with a full stop after “]”
Response : It has been done and highlighted
Due to these properties DNA is an important analytical tool in the forensic analysis[8].
Query: Line 70: Need to insert a full stop after "]".
Response : It has been done and highlighted
The sources of DNA include soft tissues such as human skin [9] dental pulp odontoblastic processes [10] and cellular cementum [11].
Query: Line 74: Need to insert a full stop after "]".
Response : It has been done and highlighted
Of all the external sources, the toothbrush is a reliable source of DNA that is easy to procure[6].
Query: In table 1 should be added the column “Capillary Electrophoresis Instrument” after the
column "Gene(s) assessed with method".
Authors should describe the amplifying method in the column "Gene (s) assessed whit
method". In contrast, in the column "Capillary Electrophoresis Instrument", the authors
could report the capillary electrophoresis method used for STRs separation.
In table 1, in the column "Results", authors need to insert DNA concentration unit (e.g. ng/μg
or pg/μg
Response: The necessary changes have been made and highlighted in table 1.
Reviewer 2:
It is an interesting article dealing with an important from medicolegal point of view, practical problem of the usefulness of toothbrush as a source of DNA for forensic purposes.
My suggestions for the authors:
Query: in the line 67 please add appropriate reference literature;
Response: The appropriate reference has been added
Query: please check the punctuation again;
Response: The corrections have been made and highlighted.
Query: there are some typos! – check it;
Response: The corrections have been made and highlighted.
Query: Try to answer which part of the toothbrush is the best source of DNA; is there any data about comparison of different parts of toothbrush? If not, write it in the main text.
Response: Since there is lack of sufficient information the below information has been included in the discussion
Also, there is lack of information on which part of the tooth brush is a good source of DNA. Hence future studies need to report the chemical composition of bristles of the tooth brush. Comparative studies on the DNA yield with different chemical compositions and DNA yield with different tooth brush parts could also be conducted.
Reviewer 3:
Query: Title and abstract needs revision as suggested in file.
Response: the suggestions have been incorporated and highlighted
Query: Major body of article needs revision as suggested in attached file under track changes.
Response: The suggestions have been incorporated and highlighted.
Query: The review did relate to the title, however, I feel the title should be reworded to "Toothbrushes as a source of DNA for gender and human identification - A systematic review"
Response : The title has been changed and highlighted
Toothbrushes as a source of DNA for gender and human identification - A systematic review
Query: As a review it is relevant, although the total number of papers included was small. As a review, it summerises the literature in this topic. The authors could have increased the scope in the literature search, yet they did initiate a broad search. In writing the introduction , they could have mentioned that there are some negatives in DNA analysis, in that there needs to be absolute certainty that the toothbrush belonged to the person being investigated, that the toothbrush has not been contaminated by others including members of the household and there was proper protocol in collecting, transporting and analysis of the evidence.
Response: The same has been included in the introduction and highlighted.
There are numerous factors that influence the successful collection and extraction of DNA. They are duration of use of toothbrushes, the need for absolute certainty that the toothbrush belonged to the person being investigated, that the toothbrush has not been contaminated by others including members of the household, proper protocol in collection, transportation, to obtain cellular material from the used toothbrush method, DNA extraction and analysis of the evidence.
Query: The conclusions are adequate and address the issues. However the chemical composition should have been mentioned in the discussion rather than the conclusion.
Response: As advised, the sentence on composition is shifted to the discussion as follows:
One of the major limitations of the included studies is that none of the studies mentioned the chemical composition of the bristles which could also affect cell adhesion and therefore could alter DNA yield[17,18,24–26].
Query: References were appropiate. Figure 1. was appropiate, yet Table 1columns were too narrow in 5,6,and 7 and I suggest it would be better to seperate Table 1 into two tables to create more space so it would be easier to read.
Response: Table 1 is placed in a landscape page thus providing sufficient width for each column to allow easier read. MDPI often allows such landscape-based pages to accommodate the broader tables.
Query: Table 2 seems unnecessary as the bias was clearly mentioned in the body of the paper.
Response: Table 2 provides a detailed assessment of what parameters were present in each of the included studies. While the write up on risk of bias just summarizes these parameters as a whole. Thus, without the table 2, readers would not know the reason for stratifying each study with the allotted levels of risk of bias.

Reviewer 2 Report
It is an interesting article dealing with an important from medicolegal point of view, practical problem of the usefulness of toothbrush as a source of DNA for forensic purposes.
My suggestions for the authors:
- in the line 67 please add appropriate reference literature;
- please check the punctuation again;
- there are some typos! – check it;
- try to answer which part of the toothbrush is the best source of DNA; is there any data about comparison of different parts of toothbrush? If not, write it in the main text.
Author Response

(The authors gave the same response as above.)

Reviewer 3 Report
Title and abstract needs revision as suggested in file.
Major body of article needs revision as suggested in attached file under track changes.
- The review did relate to the title, however, I feel the title should be reworded to "Toothbrushes as a source of DNA for gender and human identification - A systematic review"
- As a review it is relevant, although the total number of papers included was small.
- As a review, it summerises the literature in this topic.
- The authors could have increased the scope in the literature search, yet they did initiate a broad search. In writing the introduction , they could have mentioned that there are some negatives in DNA analysis, in that there needs to be absolute certainty that the toothbrush belonged to the person being investigated, that the toothbrush has not been contaminated by others including members of the houshold and there was proper protocol in collecting, transporting and analysis of the evidence.
- The conclusions are adequate and address the issues.However the chemical composition should have been mentioned in the discussion rather than the conclusion.
- References were appropiate.
- Figure 1. was appropiate, yet Table 1columns were too narrow in 5,6,and 7 and I suggest it would be better to seperate Table 1 into two tables to create more space so it would be easier to read. Table 2 seems unnecessary as the bias was clearly mentioned in the body of the paper.

Author Response

(The authors gave the same response as above.)

Reviewer 4 Report
The Authors present a review about the toothbrushes as a reliable source of DNA for person or gender identification.
The topic of the article is interesting, and it is relevant for the Forensic Medicine and research.
English is quite good.
Abstract is well written and focuses on the topic. The results are clear and concise.
Introduction is well written and gives a brief description about the topic. The methodology of the research study is correct as well.
It seems excessive to me that out of 130 articles, 120 were discarded because they were redundant. In my opinion the Authors should better explain the selection and exclusion procedure. How many duplicates were there? How many studies were considered irrelevant and why?
Tables are sufficiently exhaustive for the authors' purpose.
In my opinion, the Authors should better describe the practical purposes and limitations of the study in the closing part of the discussion.
In conclusion, the message of the Authors and the Forensic implication seem interesting, but a revision of the article is necessary.
Author Response
Reviewer 1:
This review shows that toothbrushes or siwaks are a possible source of DNA for person identification. Still, we need interlaboratory studies about DNA stability and recovery for its acceptance in court proceedings.
Query: The authors may wish to add to the systematic review a hypothesis of future
interlaboratory study about toothbrushes or siwaks storage procedures for DNA recovery.
The analysis could be conduct with a standard multiplex to unify the process and start
implementation in forensic systems.
Response: The same has been included and highlighted
Considering that tooth brushes or Siwaks are an important source of forensic evidence it would be worthwhile to form protocols for handling and storage of tooth brushes tooth brushes or Siwaks as they could aid in critical problem solving in criminal proceedings. The studies included in this systematic review have revealed significant information about recovery of DNA from toothbrushes however it is to be reiterated that only low yield of DNA could be harvested from the tooth brushes due to the fact that toothpastes contain an array of PCR inhibitors. We therefore recommend research and development towards development of dentifrices free of PCR inhibitors considering the mammoth importance of toothbrushes in the forensic view point. Moreover, storage of tooth brushes in toilets is another concerning fact as tooth brushes could be contaminated by faecal plumes which also contain PCR inhibitors and would contaminate the tooth brushes with microbial and viral DNA. Hence we recommend storage of tooth brushes in a secluded space.
Minor Issues
Query: Line 55: Words "reason behind" could be erased
Response : The sentence has been rephrased and highlighted
Human identification in such events is essential to investigate the nature of the calamity/accident, persons involved, assist the police and courts of law and give closure to relatives
The authors should review punctuation, e.g. at lines 67, 68, 70, 74
Query: Line 67: Need to replace the v with a full stop after the word stability
Response : It has been done and highlighted
It has a double-stranded structure and is one of the most stable natural molecules that can be preserved for millions of years without losing structure and stability.
Query: Line 68: Need to replace the “(“with a full stop after “]”
Response : It has been done and highlighted
Due to these properties DNA is an important analytical tool in the forensic analysis[8].
Query: Line 70: Need to insert a full stop after "]".
Response : It has been done and highlighted
The sources of DNA include soft tissues such as human skin [9] dental pulp odontoblastic processes [10] and cellular cementum [11].
Query: Line 74: Need to insert a full stop after "]".
Response : It has been done and highlighted
Of all the external sources, the toothbrush is a reliable source of DNA that is easy to procure[6].
Query: In table 1 should be added the column “Capillary Electrophoresis Instrument” after the
column "Gene(s) assessed with method".
Authors should describe the amplifying method in the column "Gene (s) assessed whit
method". In contrast, in the column "Capillary Electrophoresis Instrument", the authors
could report the capillary electrophoresis method used for STRs separation.
In table 1, in the column "Results", authors need to insert DNA concentration unit (e.g. ng/μg
or pg/μg
Response: The necessary changes have been made and highlighted in table 1.
Reviewer 2:
It is an interesting article dealing with an important from medicolegal point of view, practical problem of the usefulness of toothbrush as a source of DNA for forensic purposes.
My suggestions for the authors:
Query: in the line 67 please add appropriate reference literature;
Response: The appropriate reference has been added
Query: please check the punctuation again;
Response: The corrections have been made and highlighted.
Query: there are some typos! – check it;
Response: The corrections have been made and highlighted.
Query: Try to answer which part of the toothbrush is the best source of DNA; is there any data about comparison of different parts of toothbrush? If not, write it in the main text.
Response: Since there is lack of sufficient information the below information has been included in the discussion
Also, there is lack of information on which part of the tooth brush is a good source of DNA. Hence future studies need to report the chemical composition of bristles of the tooth brush. Comparative studies on the DNA yield with different chemical compositions and DNA yield with different tooth brush parts could also be conducted.
Reviewer 3:
Query: Title and abstract needs revision as suggested in file.
Response: the suggestions have been incorporated and highlighted
Query: Major body of article needs revision as suggested in attached file under track changes.
Response: The suggestions have been incorporated and highlighted.
Query: The review did relate to the title, however, I feel the title should be reworded to "Toothbrushes as a source of DNA for gender and human identification - A systematic review"
Response : The title has been changed and highlighted
Toothbrushes as a source of DNA for gender and human identification - A systematic review
Query: As a review it is relevant, although the total number of papers included was small. As a review, it summerises the literature in this topic. The authors could have increased the scope in the literature search, yet they did initiate a broad search. In writing the introduction , they could have mentioned that there are some negatives in DNA analysis, in that there needs to be absolute certainty that the toothbrush belonged to the person being investigated, that the toothbrush has not been contaminated by others including members of the household and there was proper protocol in collecting, transporting and analysis of the evidence.
Response: The same has been included in the introduction and highlighted.
There are numerous factors that influence the successful collection and extraction of DNA. They are duration of use of toothbrushes, the need for absolute certainty that the toothbrush belonged to the person being investigated, that the toothbrush has not been contaminated by others including members of the household, proper protocol in collection, transportation, to obtain cellular material from the used toothbrush method, DNA extraction and analysis of the evidence.
Query: The conclusions are adequate and address the issues. However the chemical composition should have been mentioned in the discussion rather than the conclusion.
Response: As advised, the sentence on composition is shifted to the discussion as follows:
One of the major limitations of the included studies is that none of the studies mentioned the chemical composition of the bristles which could also affect cell adhesion and therefore could alter DNA yield[17,18,24–26].
Query: References were appropiate. Figure 1. was appropiate, yet Table 1columns were too narrow in 5,6,and 7 and I suggest it would be better to seperate Table 1 into two tables to create more space so it would be easier to read.
Response: Table 1 is placed in a landscape page thus providing sufficient width for each column to allow easier read. MDPI often allows such landscape-based pages to accommodate the broader tables.
Query: Table 2 seems unnecessary as the bias was clearly mentioned in the body of the paper.
Response: Table 2 provides a detailed assessment of what parameters were present in each of the included studies. While the write up on risk of bias just summarizes these parameters as a whole. Thus, without the table 2, readers would not know the reason for stratifying each study with the allotted levels of risk of bias.
Reviewer 4:
The Authors present a review about the toothbrushes as a reliable source of DNA for person or gender identification.
The topic of the article is interesting, and it is relevant for the Forensic Medicine and research. English is quite good. Abstract is well written and focuses on the topic. The results are clear and concise. Introduction is well written and gives a brief description about the topic. The methodology of the research study is correct as well.
Response: The authors thank the reviewers for their positive comments
Query: It seems excessive to me that out of 130 articles, 120 were discarded because they were redundant. In my opinion the Authors should better explain the selection and exclusion procedure. How many duplicates were there? How many studies were considered irrelevant and why?
Response: The suggested information has been incorporated and highlighted. The search strategy has been included as supplementary file for reference.
Out of the 60 articles in Pubmed only 4 were relevant to the study and the rest of the articles had not aimed to assess human DNA from tooth brushes. Out of the 27 articles from Scopus only one article was relevant, 10 were duplicates and the remaining 16 were irrelevant to the topic of interest. Out of the 37 articles from Web of Science one was relevant, 10 were duplicates and remaining 26 were not relevant to the topic if interest. The reason for so many articles not relevant to the topic of interest may be attributed to the decreased number of articles in the present topic. Two articles were obtained from cross reference of the relevant articles. Following the full-text screening of 8 articles only five articles met the eligibility criteria and were included in this review. Two articles were rejected as they assessed the buccal mucosal cells that were obtained as a result of brush biopsy and one was a case report.
Query: Tables are sufficiently exhaustive for the authors' purpose.
Response: The authors thank the reviewers for their positive comments.
Query: In my opinion, the Authors should better describe the practical purposes and limitations of the study in the closing part of the discussion.
Response: The changes have been made and highlighted
Considering that tooth brushes or Siwaks are an important source of forensic evidence it would be worthwhile to form protocols for handling and storage of tooth brushes tooth brushes or Siwaks as they could aid in critical problem solving in criminal proceedings. The studies included in this systematic review have revealed significant information about recovery of DNA from toothbrushes however it is to be reiterated that only low yield of DNA could be harvested from the tooth brushes due to the fact that toothpastes contain an array of PCR inhibitors. We therefore recommend research and development towards development of dentifrices free of PCR inhibitors considering the mammoth importance of toothbrushes in the forensic view point. Moreover, storage of tooth brushes in toilets is another concerning fact as tooth brushes could be contaminated by faecal plumes which also contain PCR inhibitors and would contaminate the tooth brushes with microbial and viral DNA. Hence we recommend storage of tooth brushes in a secluded space.
One of the major limitations of the included studies is that none of the studies mentioned the chemical composition of the bristles which could also affect cell adhesion and therefore could alter DNA yield[17,18,24–26]. Also, there is lack of information on which part of the tooth brush is a good source of DNA. Hence future studies need to report the chemical composition of bristles of the tooth brush. Comparative studies on the DNA yield with different chemical compositions and DNA yield with different tooth brush parts could also be conducted.
Query: In conclusion, the message of the Authors and the Forensic implication seem interesting, but a revision of the article is necessary.
Response: The suggested changes have been incorporated and highlighted.
